# Experimental Study on Shared Bike Use Behavior under Bounded Rational Theory and Credit Supervision Mechanism

**Yao Yao [1], Linwei Liu [2], Zibin Guo [1], Ziheng Liu [1] and Huiyu Zhou [1,*]**

1   School of Economics and Management, Beijing Jiao Tong University, Haidian District, Beijing 100044, China; 16241148@bjtu.edu.cn (Y.Y.); 16241124@bjtu.edu.cn (Z.G.); 14241089@bjtu.edu.cn (Z.L.)
2   School of Traffic and Transportation, Beijing Jiao Tong University, Haidian District, Beijing 100044, China; 16251042@bjtu.edu.cn
*   Correspondence: hyzhou@bjtu.edu.cn; Tel.: +86-10-5168-3773

**Abstract:** As a new travel model, the bike-sharing system (BSS) solves the 'last kilometer' problem and has developed rapidly for its convenience. However, many accompanying problems have emerged. In China, parking violation problems—such as severe traffic congestion—are caused by dock-less shared bikes. Furthermore, a large number of shared bikes have to be scrapped early for vandalism. As a special form of public good, bike-sharing also faces the dilemma of negative externalities. Seeking a solution, Mobike has conducted a credit supervision mechanism, which transfers the users' different behavior to credits for user behavior regulation, but with unsatisfactory results. The goal of the paper is to test the validity of credit supervision mechanism from user's perspective to regulate the abuse of sharing bike by simulating the use scenario of BSS in real life in a lab experiment based on induced value theory. The behavioral and pre- and post-experiment survey data were thoroughly analyzed. The results show that, within a negative context, the credit supervision system has a more significant effect on inducing proper user behavior, which improves after adding a real-time feedback mechanism. Finally, we provide effective suggestions to policy makers and shared bike companies for inducing positive user behavior.

**Keywords:** bike-sharing; credit mechanism; experimental economics; externality; induced value theory; bounded rationality

## 1. Introduction

As a new type of green travel mode, the trend of shared-bikes has swept across urban areas. Its convenience, environmental-friendliness, cost-efficient price, and the ability to connect with other modes of transportation is attracting more and more attention. In December 2017, the number of shared bicycle users in China has reached 221 million. The user scale has increased by 115 million within half a year with a growth rate of 108.1%. Furthermore, in 2018, the number of shared-bike users will reach 298 million [1]. However, with the wild growth of many bike-sharing companies and the rapid increase in the number of shared-bikes, chaos has begun to emerge. Although government officers and company managers are trying to regulate shared-bike users' behavior via policy and a credit system, the current result is still unsatisfactory.

Therefore, this paper aims to tackle shared bike users' abuse problem, and provide implications for more effective regulation mechanism—e.g., credit supervision mechanism—which can induce proper shared bike users' behavior, and maintain the sustainable development of this green traffic mode. Credit supervision system in sharing bike context refers to the system trying to manage the

rewards, or punishment, of users on the basis of their personal behavior by induce the proper behavior with rewards and reduce the abuse behavior by penalties.

As early as the 1960s, the first public bicycle appeared in the Netherlands [2] and attracted the attention of many researchers. This travel model first developed spontaneously in the initial stage and then later, in the second period (1991–1995), more standardized public bike systems emerged typically with designated parking docks. Shaheen [3] pointed out the role of smart bike-sharing systems in urban mobility—it solved the 'last kilometer' problem. However, during the first two years of implementation, shared-bikes suffered from loss and damage in France to a greater extent than during the trial period in Hangzhou. The third period stretched from 1996 to 2007. At this time, shared bicycles were more intelligent and convenient, with embedded GPS positioning systems, but the user growth rate was still slow. The game changer came with the rapid development of the sharing economy [4]. Different from public bicycle services in Europe and America which usually provided designated parking docks, the shared bike service in China features dock-less bikes which riders can use and park anywhere.

From 2008, led and managed by the private capital market, shared bikes emerged rapidly all over the country. The dock-less characteristic of the shared-bikes implies that riders can use and park the shared-bikes anywhere they feel convenient; therefore, the flexibility and convenience attracted many users. China's bike-sharing system has already transformed the look and feel of cities, with more than 200 million apps downloaded and billions of rides taken on millions of bikes. Furthermore, it is going global, according to a Statista report, the global bike-sharing market is estimated to grow to between seven and eight billion euros by 2021. The number of bikes in bike sharing schemes is expected to reach around 20 million units during the same time period.

However, this new mode has its own problems. Shared-bikes have the non-competitive and non-exclusive characteristics of public goods in the context of full layout, therefore, similar to all public goods [5], they face negative externality problems. Firstly, users tend to park bicycles haphazardly, infringing public space—the strength of the system also has a dark side; secondly, the ambiguity of property, because of its sharing nature, leads to more traffic violations and dangerous user behavior; thirdly, as a public facility with the nature of shared property, shared bicycles also come with the problem of being stolen, vandalized, and used roughly, causing losses exceeding the expected level. In short, shared bikes are being abused. This problem not only incurs huge costs for bike-sharing companies, but also causes other social disorder and ethical concerns.

In order to pursue the sustainable development of the BSS, regulating shared-bike users' behavior and avoiding abuse has become a top priority for the government and bike-sharing companies.

Indeed, forcing the shared-bike companies to repair and protect bikes or to manage the indiscriminately parked bikes in a timely manner may be an efficient method to solve the problem; however, it forces huge and nearly unbearable extra maintenance costs onto the shared-bike companies. On the other hand, inducing users' individual spontaneous behavior may be a more efficient and cost-friendly solution. Moreover, strong policy regulations and technical support are only a part of the solution, and more importantly, shared-bike users should cope with the complex and dynamic traffic system with self-discipline. After all, the sharing economy means that self-discipline matters [6]. The government needs to coordinate and ensure that companies induce and guide users to raise their own moral literacy and legal awareness.

At present, the operation of BSS relies almost entirely on government regulations and company management. Solutions adopted rarely tackle the problem from the users' perspective, which might be more efficient in the sharing scenario. Only a small number of BSS companies have cooperated with Sesame Credit, a kind of credit evaluation system created by Internet company Ali, which judges their credit level based on the 'sesame score' to link the behavior of bike users with their sesame scores. However, the efficiency of this credit supervision system is unclear and still has room for improvement. Therefore, it is interesting to explore why it was not actually effective with more thorough analysis from the perspective of user behavior.

On the other hand, most of the current studies about shared-bike users' behavior, for example, Chen et al.'s [7] study of users' illegal parking behavior of bicycle-sharing based on norm activation theory and Wang et al.'s [8] analysis of shared-bike users' behavior, [8] among others, utilize stated preference (SP) data and their estimations are only based on questionnaires. However, SP data just reflects people's intentions, which do not always lead to the intended actual behavior response. Therefore, the inconsistency between the stated preference results and the revealed actual behavior is difficult to detect and impairs the investigators' conclusion. In fact, there is no historical or expected behavior of causality in the assumed traffic environment [9]. Direct observation of the behavior of the respondents is a better choice, but large-scale field research is rare because of the high cost. In this case, experimental economics (EE) [10] provides a good alternative that allows direct observation of actual behavior at a lower cost, while being able to control and manipulate the traffic environment in a selected manner.

Moreover, bounded rationality [11] provides realistic theoretical assumptions of empirical methods and mathematical deduction to include irrational and uncertain factors in human behavior or economic relations in the context of shared-bikes. Therefore, this paper will explore how to induce proper user behavior, and more specifically, how to build an effective credit supervision mechanism to induce preferred user behavior, based on bounded rationality theory. It will also study the corresponding effect of the corresponding behavioral approach by carefully setting and analyzing economic lab experiments.

The contributions of this research mainly include following parts: (1) targeted verification and exploration of the effectiveness of credit supervision mechanisms in shared-bicycle behavioral scenarios; (2) the innovation of methodology to study BSS by laboratory experiment of experimental economics (EE), majority of studies of which concentrate on using questionnaire and minority on field experiment, which will be discussed in Section 2.

The detailed illustration of the innovation about the methodology is as follows:

Firstly, bounded rationality instead of traditional 'rational man' assumption is applied in the experimental design to study users' behaviors.

Secondly, based on experimental economics and behavioral economics theory, the lab experiment is designed while maintaining proper internal and external integrity of experiment [12] and combined with BSS's public goods characteristics. Therefore, the lab experiment is originally designed to imitate the scenario of using shared bikes.

Thirdly, the actual behavior of the participants can be directly observed, therefore it is the revealed preference (RP) data instead of the stated preference (SP) data, which is obtained from the questionnaires, is collected from this lab experiment. By setting up the pre- experimental and post-experimental questionnaires, the comparison of the SP data from the questionnaire with the RP data from the formal experiment can be conducted, which enables us to clearly compare the differences between the SP and RP data.

Fourthly, from the perspective of user behavior induction, four different credit supervision mechanisms are designed in the experiment in order to alleviate the negative externalities of shared bikes, and promote the green sustainable development of urban transportation.

The remainder of this paper is organized as follows. The next section reviews the literature on bike-sharing and also on pro-social behavior. Section 3 presents the research design for this study, including survey design, data collection, statistical analysis of participants, measurements of key concepts, and methodology. Descriptive analysis results are shown in Section 4. Furthermore, Tukey test and a logit model analysis is presented in Section 5 to measure the significance of potential factors and also their influence on proper shared-bike parking behavior, and the model estimation results are discussed in Section 6. The conclusion and discussion are presented in the final section.

## 2. Literature Review

### 2.1. Qualitative Research on Abuse Behavior of Shared Bike

Because of the natural characteristics of shared-bikes, it is too costly to supervise the behavior of each user everywhere. Therefore, it is particularly necessary to introduce an invisible supervision mechanism, which is also an effective solution.

The main sources of policy regulation include two subjects: government agencies and operating companies. Research on government legal supervision is mainly carried out in the context of police penalties for uncivilized use of shared bikes, but is lacking to pay attention to the differences between shared bikes and other private vehicles. The government should also define the division of bicycle responsibility clearly, to identify a number of current illegal activities involving shared-bikes. Furthermore, some researchers proposed to band several kinds of regulation together. For instance, Cherry and Cervero [13] stress combining government, enterprises, and the public to guide healthy development of the sharing system.

Some of the research refers to technological aid to tackle the problem. Razzaque et al. [14] claim to regulate shared-bike parking areas, with encouraging the use of big data information such as the scientific operational innovation management system, and making full use of the developed Internet system to promote industry self-management. Vitković [15] discussed the control of the shared bicycle market by law avenue and suggested that policy and market method with large data technology should be combined to regulate this market.

However, technology can only solve part of the problem. To better guide and regulate user behavior, a sound and effective credit supervision system should be established to increase the opportunity cost of improper behavior, so that the behavior of moral marginal groups can be effectively controlled [16]. Meanwhile, in practice, headed by Mobike and Ofo, Chinese shared-bike companies have initially established incentives and penalties for the use and parking of shared-bikes, and adopted credit points to restrict user behavior.

Current regulation systems for shared-bikes, especially the credit supervision system, mostly rely on reports from other users and thereby lack actual supervision [17]. On the other hand, the evaluation system does not reach the information sharing requirement, that is, the offender is only known to the operator. Therefore, even if the credit is deducted, the offenders might manage to choose another company, or even stop using shared-bikes [18], which limits the company's own solution to the problems [19]. Hence, Franklin et al. [20] stress that only coordinating government, enterprises and the public, together with an efficient credit supervision mechanism, can promote the development of shared-bike management in a sustainable and green direction.

To date, much research on the shared-bike credit supervision mechanism has been concentrated on policy and qualitative analysis. Limited studies have discussed and analyzed the problem of shared-bikes user abuse with a credit supervision mechanism and quantitative analysis. Therefore, this study aims to tackle this problem with both SP and RP data analysis.

### 2.2. Research on Shared-Bikes from a Behavioral Perspective

Current behavioral research on shared-bike abuse is limited. Wu et al. [21] used a field experiment research method to set up a 'survival experiment' for shared bikes to study whether people would pick up the collapsed shared bike, finding that the proportion of people who did not use bicycles but chose to pick up collapsed bicycles was 632:1. Guo et al. [22] used questionnaire surveys to explore conditions that might affect the use of shared bikes. Li et al. [23] invite interviewees to research on factors influencing user abuse of shared bicycles from users' behavioral aspect. To conclude, field experiments have the disadvantages of low observation efficiency, greater experiment cost, and fewer effective samples. While, because of the vague memory of the participants or the lack of accurate self-knowledge, there was a big gap between the SP and RP of the participants, resulting in systematic errors in data collected by questionnaire survey.

Most quantitative research on shared-bikes, from a behavioral perspective, is focused on analyzing shared-bike usage and distribution mode, rather than individual behavior in the abuse context. For example, Benarbia et al. [24] established a bicycle site real-time traffic monitoring and control system using a stochastic Petri network model in the context of a mature shared-bike system in major cities such as Paris and London in Europe, and followed up in 2013. Shu et al. [25] analyzed the real-time traffic and possible capacity of each site by analyzing BSS site data in Singapore, predicting the optimal initial number of bicycles at each launching point, and establishing a BSS bicycle efficient redistribution model. These studies use statistical analysis methods to share BSS site data in different cities as data sources, trying to find the best implementation control framework. Alvarez-Valdes et al. [26] established a bicycle redistribution algorithm to estimate the service quality of BSS products by estimating the unmet needs of each site in a certain period in the future, and the number of possible bicycles at the beginning of the period.

The majority of research on shared-bikes from a behavioral perspective is focused on describing usage models, with limited literature discussing the shared-bike abuse problem. Meanwhile, the theoretical basis of these two aspects is bounded rationality. Compared with the traditional hypothesis of the rational man, bounded rationality can guide us to standardize user behavior [27].

In addition, most research has studied the macroeconomic role of shared bicycle systems in urban development instead of the individual behavioral perspective. Most of them use empirical analysis to prove positive effects, and strive to establish a more positive role for shared-bikes in green city construction. For example, Hamilton and Wichman [28] set up Stochastic network flow model to perform an exploration of sharing-bike's contribution on city sustainability construction, which concluded that sharing-bike reduces traffic congestion upwards of 4% within a neighborhood. Faghih-Imani [29] established a parameter estimation model to promote the use of shared-bicycles and adjustment factors with data from Barcelona and Sevill. Lan et al. [30] explored factors in the shared economy using a mixed case approach to transform people from passive product service recipients to active value co-creators. Finally, it was found that cognitive and time-value factors are effective and can have a big influence on people's participation in value creation and their creative willingness. Bullock et al. [31] concentrated on effective factors on sustainability of BSS from urban traffic planning, system design, and business model perspectives.

### 2.3. Literature of Experimental Economics

EE, on the other hand, provides us with another perspective to imitate real user behavior within an experimental environment. Therefore, experiments can be tested with multiple runs until the policy settings and parameters actually meet the requirements. Unlike the real world, where policies are either 'conducted' or not, the experimental context is much more flexible and convenient for exploring the combined effect of different policy-mixtures. Many social science studies have used EE methodology. For example, Pluntke and Prabhakar [32] conducted a field experiment in Singapore with the aim of shifting public transit users from one mode to another. They found that those who became members through friends' invitations, and those who chose the raffle reward over the fixed reward, shifted more than others. Tørnblad et al. [33] implemented a field experiment to test whether information about public transit options, alone or in combination with a small set of free bus tickets, would shift travel choice from cars to buses. Furthermore, those who chose the raffle reward over the fixed reward, shifted more than others from on-peak to off-peak travel. Hultkrantz and Lindberg [34] conducted a field experiment of a pay-as-you-speed insurance scheme involving 95 drivers in Sweden. Dixit [35] used a laboratory experiment to study route choice tasks in driving simulators to estimate risk attitudes and compared these with risk attitudes estimated from more standard stylized lottery choices in 2014. Compared with questionnaire surveys, researchers could more precisely collect participants' choice preferences and behavior.

The same can be applied to the research on shared-bike abuse from a behavioral perspective. In the context of trying to control and supervise the shared-bike abuse scenario, EE can be a powerful

tool to solve problems. More specifically, EE can be utilized to study the different and combined effects of user behavioral response to different credit supervision mechanisms. To this end, (EE) approaches adopted in this study provide an alternative approach that allows direct observation of actual behavioral response at a lower cost, at the same time controlling and manipulating the traffic environment in a chosen manner.

*2.4. Comparison*

Therefore, as shown in Table 1, compared with other literature, the features of this research are: firstly, to explore abusive use behavior of the shared-bike market with quantitative models. Secondly, the study applied EE approaches of laboratory experiments to avoid the impact of declarative preferences on data acquisition. Compared with SP data, the real behavior revealed in the experiment can be used for repeatable and more accurate behavioral analysis. Thirdly, the paper sets up a proper laboratory experiment paradigm regarding the shared-bike abuse scenario. Compared with field experiments, it is possible to directly ensure the internal consistency of the entire experimental process through simulation and ingenious experimental framework [10]. The experimental process can be repeated and controlled strictly, and is potentially re-usable for other similar contexts related to negative externalities. Fourthly, this paper tackles the abuse problem from the perspective of user behavior with efficient credit supervision mechanism, and the mechanism design relies on the bounded rationality theory according to the results of the EE experiments. The experiment evaluates if the existence of the shared-bike credit supervision system is effective and finally provides useful suggestions for the improvement of the supervision mechanism.

**Table 1.** Related literature comparison.

| Research Category | Literature | Research Focus | Analysis Method | Highlights |
|---|---|---|---|---|
| **Geographical distribution** | Sarkar et al. [36] | Penetration rate of shared bicycles | Descriptive statistics | Shared bicycles' penetration rate is improving all over the world |
| **Technology improvement** | Vitković [15] | Control of the shared bicycle market by legal avenues | Bicycle distribution algorithm | Combining policy and market method with large data technology to improve service |
| | Razzaque et al. [14] | Management of shared bicycles | Big Data approach | Set shared database to promote self-management |
| **Mixed regulation** | Cherry and Cervero [13] | Policy regulation of shared bicycles | Case study and qualitative analysis | Combining government, enterprises, and the public to guide healthy development of the sharing system |
| **Abuse behaviors** | Li et al. [23] | Factors influencing the problem of user abuse of shared bicycles | Logit regression method | Taking advantage, moral education, property rights, feeling, emergency, consumption concept, evasion |
| | Wu [21] | People's reaction to different kinds of shared bicycles | Field Experiment and descriptive analysis | Percentage of people picking bicycles in order is 99.842% |
| **Bicycle allocation** | Benarbia et al. [24] | Monitor and control of real-time flow of stations | Stochastic Petri network | Building a real time traffic monitoring and control system for bicycle stations |
| | Shu and Mabel [20] | Efficient redistribution of shared bicycles | Descriptive statistics | Establish efficient redistribution model of the shared bicycle system |
| | Ramon Alvarez-Valdes et al. [26] | Optimizing service quality of shared bicycles | Network model | Establish an efficient bicycle reassignment model |
| **Sustainable development of bike sharing system** | Hamilton and Wichman [28] | Contribution of BSS to reducing congestion | Stochastic network flow model | Bikeshare reduces traffic congestion upwards of 4% within a neighborhood |
| | Bullock et al. [31] | Sustainability of BSS from urban traffic planning, system design, and business model perspectives | Case study and descriptive analysis | Clarify the economic contribution of public bike-share to the sustainability and efficient functioning of cities. |
| | Ahmadreza Faghih-Imani et al. [29] | Factors of the use and adjustment of shared bicycles | Linear mixed model | Station density, average bearing capacity, diversity of commercial land use, number of entertainment places and restaurants in the related sub-urban area |
| | Lan [30] | Factors changing people's value of shared bike services | Case study and descriptive Analysis | Cognitive and time value factors are important |
| **Shared bike credit supervision system** | This study | Effectiveness of shared bicycle credit supervision mechanism | Logit regression and lab experiment study | Only the credit supervision mechanism significantly effectual, while personality, gender, and previous experience of using shared bicycles ineffectual on user behavior |
| | Chen [16] | Chaos control of shared bicycle | Case study | Increase the cost of mistakes by credit system to effectively control users' behavior |

## 3. Experimental Design

This study will utilize bounded rationality theory from a behavioral perspective to study the abusive behavior of shared bicycle users, and conduct experiments to test the impact of a credit supervision system on user behavior, thus providing reasonable suggestions for the improvement of the shared bicycle credit system.

### 3.1. Experimental Process Design

This experiment can be divided into three parts: the preliminary experiment, formal experiment, and follow-up experiment. In the pre-experiment, participants were required to fill out a pre-experiment questionnaire and complete personality tests; thus, the basic information of participants was collected and further utilized for the grouping process.

The experimental participants were divided into five groups, one control group and four treatment groups. To ensure an unbiased process and eliminate other influences, all the participants could try several rounds of exercises; therefore, they understood the whole process, eliminating the influence of an unfamiliar environment. The true purpose of the experiments is hidden behind an outdoor game activity. At the beginning of the formal experiment, the participants in the treatment groups were given full information about how to get scores in the game and the possible consequences of violation of the game rules. The participants in the control group were only informed of the ground rules without any inducement information, and they could judge some scenarios according to their own wishes.

In the experiment, four points—A, B, C, and D—were set up and one staff member was responsible for recording and supervising the students' experiment performance at each point. Extra patrol staff monitored the participants' behavior in real time during the experiment.

As shown in Figure 1, after the behavioral experiment, participants in each treatment group filled in a post-test questionnaire. The questionnaire mainly included questions about the participants' opinions on the experiment, their feelings on the credit scoring mechanism in the experiment, and their real feelings on the credit scoring mechanism, so as to determine whether the credit supervision system had an impact on their behavior and decision-making in the experiment.

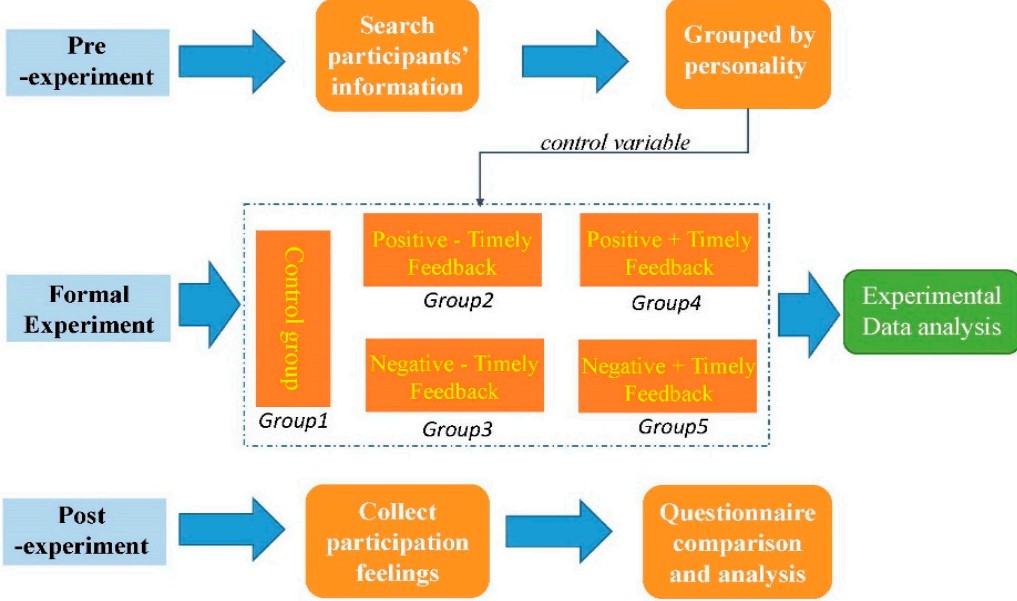

**Figure 1.** Experimental process.

### 3.1.1. Pre-Experiment Process

First of all, we designed questionnaires and invited participants to fill them in, thereby collecting the basic social demographic information of participants, daily usage of shared-bikes, travel patterns, and personality traits.

Secondly, to investigate the effect of personality traits on riding behavior, this paper adopted the Enneagram model [37] to cluster the participants' personality and divided them into three categories: thinking-oriented (Types 5–7) with thought as the original driving force, emotional dominance (Type 2–4) with emotion as the original driving force, and instinctive dominance (Type 8,9, Type 1) with action as the original driving force [38]. All three categories of participants were equally allocated in the control group and four treatment groups to ensure that the participants' personality traits did not interfere with the experiment results. Personality testing and control variables are mainly based on the theory of internal validity [39], which refers to the prediction of experimental results in accordance with the theoretical basis of experimental design. It depends on the correctness of technical details, such as the random allocation of experimental subjects and the maintenance of non-processing. Conditions remain unchanged, ensuring accurate measurement.

Meanwhile, the external validity corresponds to the internal validity [40]. Whether a design exists or not and in the basic scope of the theory, the inductive judgment from the experimental results depends on its external validity. Theoretically speaking, therefore, external validity can be guaranteed in a laboratory environment. The laboratory implements a controlled environment in which controllable factors can be manipulated while other factors remain unchanged, and disturbances can be completely eliminated. For example, economic experiments usually try to keep the characteristics of the subjects unchanged by randomly allocated processing and completely excluding communication between the subjects. Therefore, in our experiments, we focused on making sure that the participants we recruited did not know each other, and set up staff to supervise the whole field to prohibit communication.

Through the above background investigation and personality test, the participants were classified to ensure that the background information of the treatment group and the control group was the same at the formal test, and other unrelated variables affecting the treatment group and the control group were excluded, which fully guaranteed the internal and external validity of the experiment.

### 3.1.2. Formal Experiment Process Design

After maintaining the internal integrity of the experiment, there is a natural concern that whether the external integrity can be equally guaranteed. The lab experiment should identically simulate the scenario of user behavior and concentrate on the kernel issue in the scenario setting here: the representative of the sharing bikes should capture their 'public good' features. Therefore, the formal experiment will use cardboard and scissor mimics the sharing nature of the bikes.

This experiment uses a piece of cardboard and a pair of scissors to simulate a shared-bike. Twenty equal-sized circles were printed on pieces of cardboard (as shown in Figure 2) indicating a shared-bike waiting to be used. In the experiment, the participants were required to cut out circles from the cardboard with a pair of scissors. The cutting out of one circle symbolizes one use of that shared-bike. The continuous cutting out of the circles simulated the real-life scenario of shared-bike depreciation—the cardboard underwent gradual wear and tear during times of 'use', and when it ran out of circles, it could not be 'used' any longer and the 'shared-bike' was then scrapped. Furthermore, while the cutting out of one circle with a pair of scissors successfully means normal usage of the bike, to the contrary, violations can happen in various forms. Concretely, if the participant accidently cuts into another circle, or more than one circle, it will mean too much damage to the shared-bike; if the cardboard is destroyed, it will mean the whole shared-bike is damaged and scrapped. Note that without scissors, it is impossible to cut the circle out of the cardboard, which is regarded as being unable to 'use' the shared-bike any more. Therefore, the scissors actually simulate the role of

the shared-bike lock. If the participant loses or hides the scissors, this symbolizes the behavior of damaging the lock of the shared-bike so that others cannot 'use' it.

This experimental scenario of a cardboard and scissors game vividly imitates the process of wear and tear on shared-bikes in the real world. While successfully cutting out a circle indicates the natural use of a shared-bike, other behavior all represents a certain abuse of the 'shared-bike'. The scissors are also a necessary link in the process, even if the participant has the cardboard, without a pair of scissors, no one can actually 'use' it, therefore scissors represent the 'locks' of the bikes. In the field, some shared-bike users intentionally break the locks of shared-bikes so they can keep the bikes for themselves, or some users forget to lock the bikes after use, causing theft problems. Therefore, the experimental design in this study intentionally included a separate 'lock' to study this abuse behavior.

Based on Binmore's experimental design theory [41], we simplify the operation before the formal experiment and give the participants enough time to learn, so that they can adapt to the experimental scenario before the formal test, ensuring that the experiment results are unbiased.

When the game starts, those who get pieces of cardboard and scissors are recognized as having a shared-bike and can go directly to the destination, simulating a real-life situation where cycling is more efficient than walking. If a participant did not get a piece of cardboard and a pair of scissors that represent the shared-bike, he/she has to take a long way to the destination to simulate the slower mode of walking in real life.

At the same time, if there is cheating in the experiment, such as using private scissors, hiding scissors for themselves, losing the scissors, not 'parking' the cardboard and scissors properly, not cutting the circle according to the rules, and damaging the cardboard, among others, this is considered as shared-bike abuse.

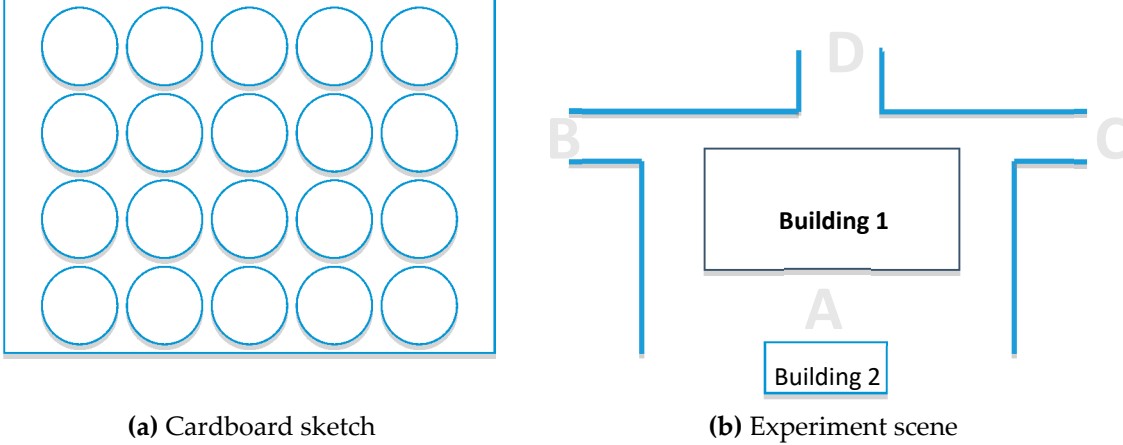

**(a)** Cardboard sketch        **(b)** Experiment scene

**Figure 2.** Cardboard sketch and the deployment of experiment scene.

The experiment set points A, B, and C as test sites, and point D as the transfer site; setting three origin points can avoid the peer effect impact on the behavior of other participants [42]. At the same time, according to the theory of external validity, we ensure that the distance between points A, B, C, and D is equal.

At the beginning of the experiment, all participants in each group were randomly assigned to points A, B, and C, moving from these points to other points. There are equal numbered participants at each starting point, with two pairs of scissors and two sheets of cardboards. Participants who get scissors and cardboard are regarded as having the shared-bike and are free to directly go to the other two experimental sites. Participants who only get scissors or just take cardboard cannot use them since they are 'broken' or 'dead-locked' sharing bikes. They can only go to the destinations without 'sharing bikes'—by 'walking'. At the recovery point D, the scissors and cardboards will be recovered by staff, and returned to the original location, imitating the maintenance process of sharing bikes.

Each participant with a 'shared bike' needs to cut a circle off the cardboard with scissor after arriving at the next experimental site imitating the natural wear and tear process of the bicycle. After cutting, another shared bike can be found at the experimental site for further testing. If they cannot find another 'shared bike', they also need to reach the intermediate site D and stay for five minutes before they go to the next destination, imitating the process of 'walking'. Intuitively, the longer travel time of 'walking' is ensured by the five minutes waiting at site D, therefore, the value of time is induced during the process. For 'walking' participants without 'sharing bike', he or he can go to the next place only after getting the seal issued by D site.

In this experiment, there were nine participants in each group and there were five groups in all. Furthermore, only one group participated at a time to further avoid peer effect. Five groups included one control group and four treatment groups (as shown in Table 2). The treatment groups were further divided into a positive context group without real-time feedback, negative context group without real-time feedback, positive context group with real-time feedback, and a negative context group with real-time feedback. The positive context group gives credit scoring mechanism to the positive context while the negative context group gives credit scoring mechanism to the negative context.

**Table 2.** Control and treatment group settings.

| | Group 1 | Group 2 | Group 3 | Group 4 | Group 5 |
|---|---|---|---|---|---|
| | **Control Group** | **Group without Real-Time Feedback** | | **Group with Real-Time Feedback** | |
| | | **Positive Context** | **Negative Context** | **Positive Context** | **Negative Context** |
| Pre-experiment | | Giving positive scoring mechanism | Giving negative scoring mechanism | Giving positive scoring mechanism | Giving negative scoring mechanism |
| A site | Observation and recording | Observation and recording | Observation and recording | Observation, recording, and inform | Observation, recording, and inform |
| B site | Observation and recording | Observation and recording | Observation and recording | Observation, recording, and inform | Observation, recording, and inform |
| C site | Observation and recording | Observation and recording | Observation and recording | Observation, recording and inform | Observation, recording, and inform |

The positive contextual group adopts the positive discourse interpretation credit scoring mechanism, while the negative contextual group uses the negative discourse interpretation credit scoring mechanism. The credit scoring mechanism in the positive and negative context is shown in Tables 3 and 4. In the two groups of real-time feedback, the staff also strictly used positive or negative context to remind participants to add or subtract credit points.

**Table 3.** Credit scoring mechanism of positive context.

| Scenarios | Score |
|---|---|
| Initial score | 100 points |
| Each trip is completed | +10 points |
| Success with cutting out the circle | +5 points |
| No violations on the trip | +10 points |

The framing effects are adopted in the treatment group scenario setting to test positive and negative contexts [27]. According to the framing effect, negative or positive framework context actually leads to different value assessment results and decision-making processes.

Therefore, based on the framing effect and prospect theory, the proposed experiment utilized the positive and negative context to simulate the effect of a credit system on the behavior choice of shared-bike users.

**Table 4.** Credit scoring mechanism of negative context.

| Scenarios | Score |
| --- | --- |
| Initial score | 100 points |
| Each trip is completed | +10 points |
| Failed to cut out the circle | −5 points |
| Other violations in the trip | −10 points |

On the other hand, the effect of timely feedback in the credit system is also studied to explore effective ways of encouraging proper behavior. The experiment is designed based on the information feedback theory from the bounded rationality perspective. Many studies show that information feedback is an indispensable factor in people's learning behavior. Ericsson et al. [43] found that adding fuel-saving route information to the new vehicle navigation system could help reduce $CO_2$ emissions during vehicle travel. Timely feedback produces the expected effective impact. Therefore, to maximize the impact of credit scores, timely feedback and non-feedback are set up in the treatment groups based on whether to inform the participants of the increase or decrease of scores when the behavior occurs.

At the same time, the experiment set up a reward mechanism, through a certain integral mechanism to assign scores to the behavior of the participant, and ultimately provide a monetary reward according to the ranking of the highest score to encourage each participant to complete as many experimental rounds as possible. Each participant will receive a 10-yuan allowance, and an extra bonus will be provided according to the corresponding scores. Monetary incentives are utilized to internalize and induce the values of time efficiency, proper behavior, and convenience. Participants are encouraged to act according to the credit scoring mechanism designed by the experiment, and participants are expected to show different behaviors in different mechanisms [44].

After the experiment, the difference between the control group and treatment groups were studied. The control group shows natural behavior without any credit mechanism where participants do not really care whether they have damaged and abused the 'shared-bike', their only goal is to complete as many rounds of the routes as possible. Therefore, the time efficiency when people use the shared-bike in the field (value of time, VOT) is simulated as the final bonus based on the scores in the control group. While in the treatment groups, participants have to trade-off between completing as many rounds of the routes as possible without abusing the 'shared-bikes'. Since violation of the rules requires a price, how they trade-off can be clearly observed during the process.

It is important to note that interaction with participants requires confidentiality of the experimental intention throughout the experiment, careful explanation of the rules, and avoidance of suggestive terminology to prevent the subject from judging whether the behavior is right or wrong before the experiment, thereby affecting the true performance in subsequent behavioral experiments [45]. Therefore, we only explained the rules before the experiment, and paid attention to the design of the pre-test to avoid direct inquiries about the shared-bicycle credit mechanism. In the formal experiment, we only guided the participants to pursue high scores, and let the participants believe that the purpose of the experiment was to seek high scores. In fact, what we were really interested in was the damage rate, cheating rate, and simulated abuse behaviors represented in the experiments.

## 4. Descriptive Analysis of Experimental Results Analysis

### 4.1. Comparison of Stated Preference (SP) and Revealed Preference (RP) Data

In order to show how revealed preferences are better predictors of behavior with respect to stated preferences, we have measured participants' feelings after experiment through three questions included in our post-experiment questionnaire for treatment group users.

The first one is "Did you think you became more careful when cutting the paperboard under the credit supervision mechanism?" with the options from "Very careful", to" Not careful at all" using five-point Likert scale. Then the following question is "Did you feel the psychological pressure of

rewards and punishments mechanism in the experiment?" with the options from" Very stressful", to "Not stressful at all" using a similar Likert scale. Finally and particularly, question 3 is "Did you think you became more careful when cutting the paperboard under the real-time feedback mechanism?" is surveyed for feedback groups only. All the analysis results of these three questions all point to one direction, and the description will be similar and repeated, so the question 2 is pulled out to illustrate as an example to discuss the SP and RP difference. No matter how they say that they do not care about the supervision system and "did not feel the stress at all", the users' unconscious behavior will show their true responds: they will decrease their abuse behavior under the carefully designed credit system.

Moreover, the reason why it is after the formal experiment is that this questionnaire about user's preference should be conducted after the experiment to avoid emitting the experimental intention, therefore the result will be unbiased [45].

### 4.1.1. Did You Feel the Psychological Pressure of Rewards and Punishments in the Experiment?

When the participants were asked "Did you feel the psychological pressure of rewards and punishments in the experiment?", out of all the participants who completed the post-test, only 8% said they felt very stressed, 24% participants selected relatively stressed, 39% of them chose almost no pressure, and 17% stated to be completely stress-free with 14% of participants expressing that their exact feeling is unknown, as shown in Figure 3.

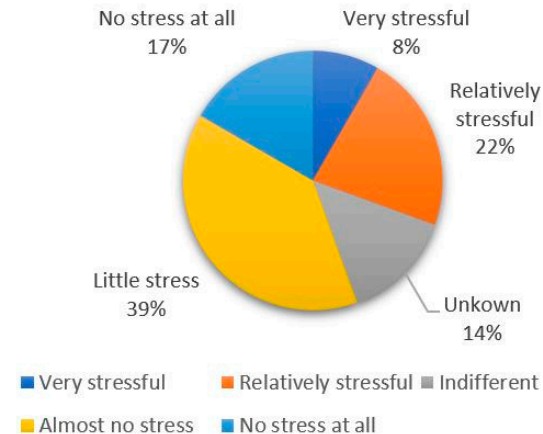

**Figure 3.** Did you feel the psychological pressure of rewards and punishments in the experiment?

At present, many traffic behavior analysis methods mainly use stated preference data (SP), that is, the participants choose by providing a variety of scenarios, a variety of options for participants, however unfortunately, participants' choices in the survey does not always match their actual choice in the field, that is, the gap exists between the stated preference and actual behavior. People often cannot fully understand the situation described in the questionnaire, so they cannot give real feedback in the SP dataset. Sometimes, they think they give real feedback. When the scenario actually happens, they will find that they themselves act alternatively. On the other hand, revealed preference (RP) data is to record the choice made by participants in real situations or experiments according to their preferences, which often reflects the actual behavior preferences of participants.

Therefore, the stated preference (SP), which is based on the results of the questionnaire, shows that more than half of the people stated that they did not felt the stress of credit supervision mechanism, only a small part of people think it is very stressful. The SP data implicated that most people believe that the credit supervision mechanism had no effect on their behavior. However, the revealed preference (RP) of these participants behavior shown by experimental data presented as Figure 4, suggested otherwise. Treatment Group 1 is the control group without any warning and incentives based on bounded rationality theory, the number of abuse behavior in Group 1 is far more than the other four treatment groups which are set under certain credit supervision mechanisms.

In Group 1, for the absence of any credit supervision mechanism, users showed limited self-discipline, by simply completing the trips as many rounds as possible. They were naturally not aware of the importance of avoiding illegal behavior. Therefore, during experiment with Group 1 involved, lots of abuse behavior is presented, we even found the losses of some 'shared bikes' during the trips.

Whereas, compared with Group 1, once the credit supervision mechanism appeared, the number of abuse behavior in Group 2 to Group 5 decreased significantly, shown as Figure 4, which proves that the effectiveness of credit supervision mechanism makes people aware of the loss of abusing shared bicycles and internalize the externality of shared bikes with value inducement design.

To conclude, the SP data, shown as the pie chart in Figure 3, participants subjectively believed they would not be affected by the advent of credit supervision mechanism, no matter what kind of the credit mechanism it was. Nevertheless, the RP data, obtained from direct observation of participants behavior in experiment and presented by Figure 4, indicated that participants in all kind of credit supervision mechanism were significantly under the shadow of the regulation of the mechanism. Compared these differences between SP and RP data, participants in this research showed an interesting behavior characteristic—although they did not recognize the impact of the credit supervision mechanism on themselves, the credit supervision mechanism did conduct an excellent constraint on participants' behavior.

Since the difference has existed to date and the way to use questionnaire (SP experiments) to collet micro-behavior data is still the main trend in various studies, the way of experimental economic which proved useful to avoid the bias of SP in questionnaire continuous to be ignored by majority of contemporary researches. This comparison further demonstrates the existence of the bias in SP data and the importance of using RP data with the help of economic experimental designing methodology.

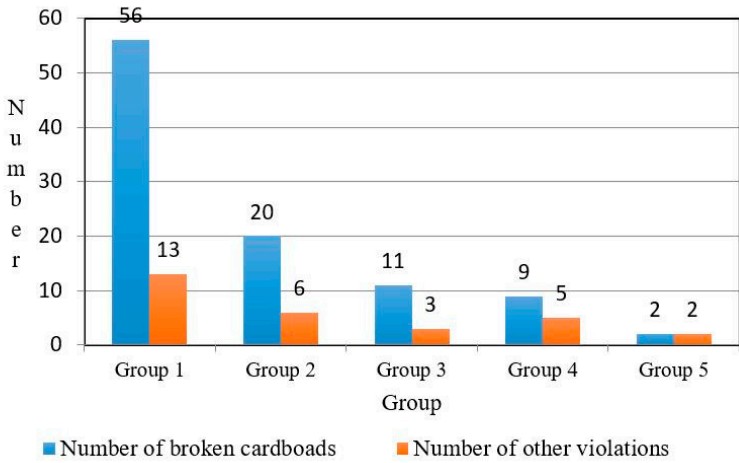

**Figure 4.** Histogram of the number of abuse behavior.

*4.2. Description Analysis about Efficiency of Different Credit Mechanism*

Credit mechanism can effectively affect users' behavior shown by the laboratory experiment in Figure 5, the number of broken cardboards in Group 1 are significantly higher than that in any other group, the same as the number of other violations. More specifically, negative context is obviously more effective than positive context when inducing people's behavior, and real-time feedback can further improve the behavior responds of participants.

In behavioral experiments, we find a very interesting phenomenon, although all treatment groups show significant behavior difference with the control group, their behavior still various according to different scenarios settings. All participants in the positive context group tend to do their best to cut the circle and then proceed to the next trip as soon as possible, even if there are minor mistakes at one time, they do not want to delay their trips too much by considering proper and cautious behavior,

since extra scores are perceived as a gain in the positive context. While, option patterns in negative contexts scenarios are completely different, participants in negative contexts prefer to take a little time and carefully trim the edges to avoid broke the circle or 'abuse' the 'shared bike'. It can be seen that participants are more sensitive to loss in the negative context, which is coincide with Tversky's prospect theory. The participants are less willing to be deducted five points for broken the cardboard during the cutting, and did not pay attention to the rule that they could add five points if they did not break it.

Furthermore, participants in the real-time feedback group spent more time to cut the circle, properly after the staff reminded them, even though they initially had an indifferent attitude. As a result, there are very few students in the real-time feedback group who have abused the 'shared bike' more than twice.

Finally, considered all four kinds of credit supervision mechanism together, efficiency of the BSS under different mechanism is measured. As shown in Figure 5, the negative feedback group (Group 5) is close to the control group on the basis of ensuring that the circle is cut, and the total number of trips of the real-time feedback group is more than that of the non-real-time feedback group. In particular, the average number of trips per person in the experimental Group 5 (i.e., the real-time feedback negative context group) was almost close to that of the experimental Group 1 (control group). As described before, the number of 'shared bike' damages and abuses of the experimental Group 5 were much smaller than that of the experimental Group 1. Therefore, the results show that the real-time feedback negative context group can reduces the social cost while maintain the similar level of travel efficiency.

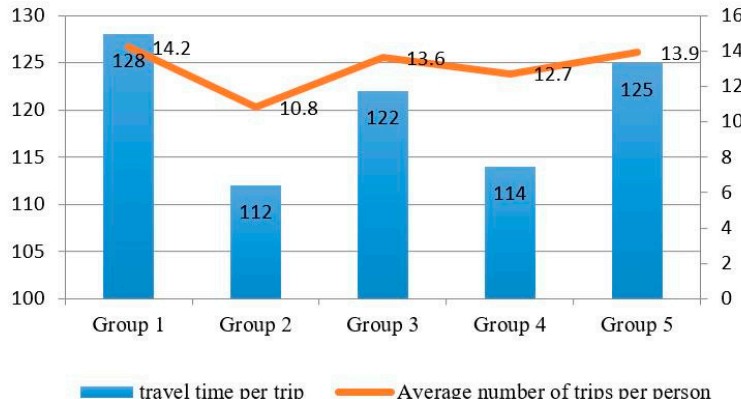

**Figure 5.** Comparison of the efficiency of experiment groups.

*4.3. Description Analysis about the Efficiency of Credit Mechanism Influenced by User Knowledge*

The credit mechanism has been introduced to Mobike shared bikes, but the effect is not significant. To investigate the reasons for this, we set the question in the post-test questionnaire, "Do you understand the sharing of bicycle in Mobike credit system?" and find one of the answers.

From the collection, the number of people commonly knew of the Mobike credit mechanism. Shown as Figure 6, 0% chose to know very well, 10% chose to know better, 7% chose not to know, 14% chose not to understand, 5% chose not to understand at all. Only 27% of the people know about the existing shared bike credit supervision mechanism, while 53% of the people do not. It can be said that the existing credit supervision system of shared bicycle enterprises is almost empty, even if users have behavior violations, they are not aware of the deducted points, or even do not know that they have engaged in abuse of the shared bicycle. Therefore, it is suggested that the real-time feedback mechanism can be adapted by the shared bike companies to inform the users their corresponding credit points in response to their behavior, and clearly inform the users of what their corresponding positive or negative incentives are.

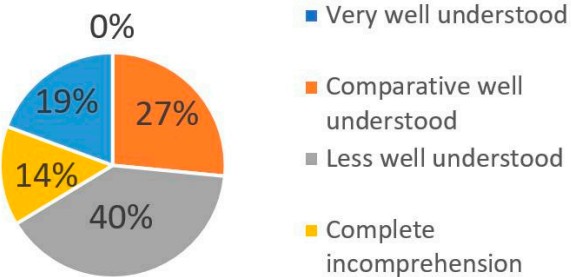

**Figure 6.** Situation of user knowledge of Mobike credit supervision mechanism.

## 5. Econometric Analysis Experimental Results

### 5.1. Damage of Cardboard (Corresponding to Damage of Shared-Bike)

The $H_0$ hypothesis was that there is no significant difference in the damage to the pieces of cardboard between treatment Groups 1–5, with the contrast hypothesis is the damage between different groups is significant. Intuitively, the alternative hypothesis is that there is a significant difference in the damage to the pieces of cardboard between treatment Groups 1–5.

The results of multiple comparison for damage by Tukey test are

**H$_0$:** $\mu_1 = \mu_2 = \mu_3 = \mu_4 = \mu_5$

**H$_1$:** *otherwise*

As shown in Turkey test result of Table 5, the mean values of Group 1 (blank control group) are significantly higher than those of Group 2 (positive context group without real-time feedback), Group 3 (negative context group without real-time feedback), Group 4 (positive context group with real-time feedback), and Group 5 (negative context group with real-time feedback), so the $H_0$ hypothesis is rejected. There are significant differences between the control group and the other four treatment groups. The credit supervision mechanism has a significant impact on the damage status of the cardboard, and therefore, effectively influences the users' behavior.

**Table 5.** Multiple comparison for damage.

| | | Dependent Variable: Damage—Tukey HSD | | |
|---|---|---|---|---|
| **(I)** | **Group** | **Difference of Mean Value (I–J)** | **Std. Error** | **Sig.(bi)** |
| **1** | 2 | 3.778 *** | 1.096 | 0.006 |
| | 3 | 5.000 *** | 1.096 | 0.000 |
| | 4 | 5.222 *** | 1.096 | 0.000 |
| | 5 | 6.000 *** | 1.096 | 0.000 |
| **2** | 1 | −3.778 *** | 1.096 | 0.006 |
| | 3 | 1.222 ** | 1.394 | 0.030 |
| | 4 | 1.444 ** | 1.810 | 0.044 |
| | 5 | 2.000 | 1.096 | 0.374 |
| **3** | 1 | −5.000 *** | 1.096 | 0.000 |
| | 2 | −1.222 ** | 1.394 | 0.030 |
| | 4 | 0.222 | 1.096 | 1.000 |
| | 5 | 1.000 *** | 1.096 | 0.009 |
| **4** | 1 | −5.222 *** | 1.096 | 0.000 |
| | 2 | −1.444 ** | 1.810 | 0.044 |
| | 3 | −0.222 | 1.096 | 1.000 |
| | 5 | 0.778 ** | 0.972 | 0.043 |
| **5** | 1 | −6.000 *** | 1.096 | 0.000 |
| | 2 | −2.000 | 1.096 | 0.374 |
| | 3 | −1.000 *** | 1.096 | 0.009 |
| | 4 | −0.778 ** | 0.972 | 0.043 |

Note: *** represents $p < 0.01$; ** represents $p < 0.05$; * represents $p < 0.1$.

Comparing Group 2 (positive contexts without real-time feedback) and Group 3 (negative contexts without real-time feedback), and Group 4 (positive contexts with real-time feedback), and Group 5 (negative contexts with real-time feedback), respectively, there were significant differences and therefore the original hypothesis is rejected. The difference between the positive context groups and negative context groups is obvious. Negative context is more effective than positive context.

Regarding real-time feedback mechanism, Group 4 (positive context group with real-time feedback), Group 2 (positive context group without real-time feedback), Group 3 (negative context group without real-time feedback), and Group 5 (negative context group with real-time feedback), respectively, are significantly different, so the original hypothesis was also rejected. There are obvious differences between groups with real-time feedback and those without. The credit supervision mechanism with real-time feedback is more binding than the credit supervision mechanism without real-time feedback.

### 5.2. Cheating Situation (Corresponding to Abuse of Shared-Bike Scenario)

The original hypothesis was that there was no significant difference in abuse rate between treatment Groups 1–5, with the contrast hypothesis is the cheating situation between different groups is significant. The alternative hypothesis is that there are significant differences in cheating among treatment Groups 1–5.

**H$_0$:** $\mu_1 = \mu_2 = \mu_3 = \mu_4 = \mu_5$

**H$_1$:** *otherwise*

The results of multiple comparison for cheating by Tukey test are

From the Tukey test result of Table 6, group1 (control group) and any other group differed significantly from each other with $p \leq 0.001$, As a result, the credit supervision mechanism successfully controlled the cheating behavior, which equals to lock, steal and other violations in using BSS. Group 2 (positive context group without real-time feedback) and Group 3 (negative context group without real-time feedback) or Group 4 (positive context group with real-time feedback) differed significantly from each other with $p \leq 0.01$, Group 3 and Group 5 (negative context group with real-time feedback) did not differ significantly from each other with $p > 0.1$, Group 4 and Group 5 did not differ significantly from each other with $p > 0.1$. As a result, there was a significant difference between the control group and the other four treatment groups.

**Table 6.** Multiple comparison for cheating.

| | | Dependent Variable: Cheat—Tukey HSD | | |
|---|---|---|---|---|
| **(I)** | **Group** | **Difference of Mean Value (I-J)** | **Std. Error** | **Sig.(bi)** |
| **1** | 2 | 0.778 *** | 0.196 | 0.003 |
| | 3 | 1.333 *** | 0.196 | 0.000 |
| | 4 | 1.344 *** | 0.191 | 0.000 |
| | 5 | 1.319 *** | 0.202 | 0.000 |
| **2** | 1 | −0.778 *** | 0.196 | 0.003 |
| | 3 | 0.556 * | 0.196 | 0.052 |
| | 4 | 0.567 ** | 0.191 | 0.038 |
| | 5 | 0.542 * | 0.202 | 0.074 |
| **3** | 1 | −1.333 *** | 0.196 | 0.000 |
| | 2 | −0.556 * | 0.196 | 0.052 |
| | 4 | 0.011 | 0.191 | 1.000 |
| | 5 | −0.014 | 0.202 | 1.000 |

**Table 6.** *Cont.*

| (I) | Group | Difference of Mean Value (I-J) | Std. Error | Sig.(bi) |
|-----|-------|-------------------------------|------------|----------|
| | 1 | −1.344 *** | 0.191 | 0.000 |
| | 2 | −0.567 ** | 0.191 | 0.038 |
| **4** | 3 | −0.011 | 0.191 | 1.000 |
| | 5 | −0.025 | 0.197 | 1.000 |
| | 1 | −1.319 *** | 0.202 | 0.000 |
| | 2 | −0.542 * | 0.202 | 0.074 |
| **5** | 3 | 0.014 | 0.202 | 1.000 |
| | 4 | 0.025 | 0.197 | 1.000 |

Note: *** represents $p < 0.01$; ** represents $p < 0.05$; * represents $p < 0.1$.

## 5.3. Universality of the Credit Supervision Mechanism

Logit regression analysis was conducted in this section on abusive behavior. (Y) from the personal perspective of participants, where Y indicates whether they engage in abusive behavior or not. The independent variables include the positive context (T1), where 0 denotes non-positive context and 1 denotes positive context; negative context (T2), where 0 denotes non-negative context and 1 denotes negative context; real-time feedback (T3), where 0 denotes no real-time feedback and 1 denotes real-time feedback; gender (SEX), where 0 denotes male and 1 denotes female; whether or not they have experience of using shared-bikes (P) "0" means do not use a shared-bike at ordinary times, 1 means do use a shared-bike at ordinary times; personality type Xi (i = 1, 2, 3), 0 means not character Xi (i = 1, 2, 3) type, 1 means character Xi (i = 1, 2, 3) type, where Xi = 1,2,3 refers to thinking-oriented, emotional dominance, and instinctive dominance personality types, respectively. The results of the regression analysis are shown in Table 7.

**Table 7.** Regression analysis results.

| | | B | S.E | Wals | df | Sig. | Exp (B) |
|---|---|---|---|---|---|---|---|
| | T1(1) | −2.511 ** | 1.297 | 3.747 | 1.000 | 0.043 | 12.320 |
| | T2(1) | −2.751 ** | 1.272 | 4.674 | 1.000 | 0.031 | 15.654 |
| | T3(1) | −2.399 ** | 1.174 | 4.172 | 1.000 | 0.041 | 11.008 |
| | Sex(1) | 0.200 | 0.917 | 0.048 | 1.000 | 0.827 | 0.819 |
| Step [a] | P(1) | 0.132 | 1.288 | 0.010 | 1.000 | 0.919 | 0.877 |
| | X1(1) | −0.910 | 1.078 | 0.712 | 1.000 | 0.399 | 2.484 |
| | X2(1) | −0.169 | 0.961 | 0.031 | 1.000 | 0.861 | 1.184 |
| | X3(1) | 0.288 | 0.654 | 0.193 | 1.000 | 0.660 | 0.750 |
| | Constant | 6.112 *** | 2.175 | 7.900 | 1.000 | 0.005 | 0.002 |

Note: [a] Input variables in step 1: T1, T2, T3, Sex, P, X1, X2, X3. *** represents $p < 0.01$; ** represents $p < 0.05$.

The regression results showed that the P values of T1 (positive context), T2 (negative context), and T3 (real-time feedback) were less than 0.05. Therefore, rejecting the original hypothesis, T1, T2, and T3 had a significant impact on cheating. Sex (gender), P (experience of using a shared-bike or not) and Xi (personality) were all significantly higher than 0.05, so the original hypothesis was not rejected. Gender, P, and Xi had no significant effect on cheating. The logit regression equation is

$$P(Y = 1) = \frac{e^{6.112 - 2.511T_1 - 2.751T_2 - 2.399T_3}}{1 + e^{6.112 - 2.511T_1 - 2.751T_2 - 2.399T_3}}$$

According to the experimental results, regardless of gender, personality, or whether they usually use a shared bike or not, the only variable that significantly affects whether they will abuse the 'shared bike' or not is the concrete setting of the credit supervision mechanism. In another words, the design

of an inducement mechanism based on bounded rationality theories. Therefore, in the process of using shared-bikes, personality, gender, previous use of shared bike experience has no significant effect on the users' behavior in the proposed lab experiment result, only the credit supervision mechanism has a clear influence on the revealed behavior.

## 6. Discussion

To conclude, the credit supervision mechanism has a significant effect on the damage status of the pieces of cardboard. Meanwhile, negative context is more efficient in discouraging the abuse of shared-bikes than positive context, and the credit supervision mechanism with real-time feedback is more efficient than the mechanism without it. Therefore, the experimental results show that—based on the bounded rationality theory—a carefully designed credit supervision mechanism can have a significant effect on the control of abusive behavior. Finally, a logistic model was estimated to study the universality of the credit supervision mechanism. The data analysis also shows that in the process of using shared-bikes, personality, gender, and previous experience of using shared-bikes have no significant impact on the behavior. In this scenario, only the credit supervision mechanism had a clear constraint on the behavior of all participants.

### 6.1. Theoretical Implications

With the wild growth of bike-sharing companies and the rapid increase in the number of shared-bikes all over the world, abusive problem has begun to emerge, yielding huge costs and negative externalities to both bike-sharing companies and society. Under such circumstances, it is highly important to study the efficient supervision mechanism which influence the users' behavior, and thus help develop policies which will effectively induce proper individual behavior. However, the current studies [22,23] of bike sharing abusive behavior are limited due to the difficulty to track and monitor the long-term usage behavior in the field, therefore, previous studies were either using qualitative analysis or based on the stated preference data.

The proposed paper fills this gap by providing an alternative experimental approach. The experiments model has been set to imitate the usage scenarios of shared-bike with multiple test runs until the supervision policy settings actually meet the requirements. Unlike the real world, where policies are either 'conducted' or not, the experimental context is much more flexible and convenient for exploring the combined effect of different policy-mixtures. This experimental approach not only imitating the usage scenarios of share-bike, but also provide a new lens to study the problem of "tragedy of the commons" [46] and have the potential to test other useful treatment mechanisms.

Secondly, regarding the effectiveness of incentives, one of the interesting conclusions is that in road traffic, the reward measure appears to be more effective in persuading people to change their behavior, which would suggest that rewarding them may be more effective than punishing them [47]. However, in the context of bike-sharing supervision system, results suggest that the negative feedback seems to be the most effective policy, which is in accordance with the psychology of prospect theory [48]. Due to the loss aversion tendency in negative context, participants try harder to avoid the loss than the positive group, this explains why negative context is more effective.

Thirdly, it is also interesting to notice the difference between SP and RP data. Although the participants did not recognize the impact of the credit supervision mechanism on themselves in the stated survey, in the revealed preference experiment, the credit supervision mechanism did conduct an excellent constraint on participants' behavior. As shown in the Section 4.1.1, SP data can clearly describe the personal intention, while RP data reveals the final response. The bias between the SP and RP data actually provides interesting implications of the obstacles between the individual intension and action, which could be further explored in the future.

Therefore, in conclusion, the proposed paper extends the insight into the effect of different credit supervision mechanisms on sharing-bike usage, and the results can provides further understanding of the research on sharing-bike abuse and other fields related to behavior nudging and inducement.

*6.2. Policy Implications*

The shared-bike abuse problem is not only creating huge maintenance costs for bike-sharing companies, but also causing serious problems for the wider society. To internalize these externalities of shared-bikes, the credit supervision system in which a reward and punishment credit could be conducted to nudge proper behavior. According to the experimental results, negative context can be an effective mechanism to regulate user behavior. For deliberate damage, tampering with QR codes, theft of shared-bikes, and other serious acts of dishonesty and abuse behavior, actual penalties or negative feedback could be conducted or even connected to the society credit rating system. A corresponding 'blacklisting' mechanism could be further adopted for severe abusive behavior.

Besides, according to the comparison of results from the questionnaire surveys before and after the formal experiment and results from the direct observation of participants' behavior in the formal experiment, the difference in stated preference and revealed preference is proved. Therefore, before the large-scale adoption of regulation policy or strategies, behavioral experiments could be a better choice to test the efficiency rather than stated preference surveys.

Real-time feedback mechanism is also effective in the credit supervision mechanism. When users incur violations, bike companies should inform the user in time, such as sending notifications on the APP, or reminding them the next time they use the service and log in to the app. Information feedback can affect the user's learning behavior, so timely notification to users of their irregular behavior can make them feel the consequences of violations more intuitively. At the same time, when the user credit score is lower than a certain score, the user's permission to use the shared-bike can be temporarily suspended.

Given the fact that most of the shared shared-bike users do not understand the existing credit system as shown in Section 4.3. Policies could be developed to promote information provision to the public, make full use of various media and forms, and engage Weibo and WeChat on social platforms, etc.

## 7. Conclusions and Future Work

Many problems have followed the rapid development of bike sharing. We believe that shared bicycles have the attributes of public goods, and therefore also face the problem of negative externalities. Therefore, the bounded rationality of behavioral economics was used to conduct experimental analysis on shared-bike users' behavior. This study simulated the use and loss of shared bicycles in real life through economic behavior experiments, and collected behavioral data for in-depth analysis.

A cardboard cutting experiment paradigm was proposed to simulate the shared-bike daily use scenarios. At the same time, the participants were surveyed before and after the experiment, and the SP data of the questionnaires were compared with the RP data of the experimental behavior. Experiment results show that the credit supervision mechanism of negative context is significantly more effective on inducing user behavior. At the same time, it is better to add real-time feedback mechanism to the credit supervision mechanism. Finally, this study explores the feedback mechanism design of shared bicycle enterprises to implement a credit supervision mechanism effectively, and through effective real-time feedback, using the network platform to publicize the shared bicycle credit. This study explored an effective credit supervision mechanism from the perspective of behavior induction to help alleviate the negative externalities of shared bicycles, and thus promote the green sustainable development of urban transportation.

For various reasons, this experimental study can be improved with these future steps. There is room for further expansion of the sample size. The experimental setting variables only discuss whether there is real-time feedback and positive and negative context, and more theories from bounded rationality could be studied at a later stage. The only independent variables were gender and personality characteristics; more comprehensive variables could be analyzed in further steps. Based on this, it is possible to follow up with a larger sample size, more complex laboratory experiments, and perform multivariate data analysis. It can also be extended to actual field experiments with the

aid of shared-bike companies in the future. Combining laboratory experiments with field experiments can achieve stronger internal and external validity of the experiment results, and therefore have the potential to be adapted to large-scale actual practice.

**Author Contributions:** Data curation, Z.L.; Formal analysis, L.L.; Investigation, Z.G.; Writing—original draft, Y.Y.; Writing—review and editing, H.Z.

**Funding:** National Natural Science Foundation of China: 61602028.

**Acknowledgments:** The study was supported by National Natural Science Foundation of China (61602028).

**Conflicts of Interest:** The authors declare no conflict of interest.

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
