# Peer review of "Experimental Study on Shared Bike Use Behavior under Bounded Rational Theory and Credit Supervision Mechanism"

_sustainability, doi:10.3390/su11010127_

Round 1
Reviewer 1 Report
The article deals with the problem of damages occurring to bikes used in a bike-sharing context. Through an experiment, the authors test the validity of a "credit supervision system" in changing people's behavior in order to reduce the damages.
Strengths: the issue is relevant, the research idea is interesting and the approach fairly original.
Weaknesses: in my opinion, the policy implications provided go far beyond the actual explanatory capability of the experiment.
Abstract: a proper way to introduce the reader to the aim of the article is to highlight very clearly, in the abstract and at the very beginning of the paper, the research question. Presently the abstract introduces the BBS and some related issues in a very general way while leaving the research question in the shadow. In the abstract, when presenting the goal of the paper, two things should be improved.
First, please be explicit, not implicit: "This paper holds... In order to deal with the problems above..." (lines 18-19). The first statement is an assumption, not a problem. I suggest using a more explicit way to immediately clarify what the article is about, e.g.: "The goal of this paper is to... Given the issues with BBS..." etc.
Second: the aim description refers to a "credit supervision system" (whch is central in the article) as if it was something the reader should already know while it is a specific contextual element of the empirical setting of this paper. Please, be explicit since the very beginning about the scope of the research and about what the "credit supervision system" is and its role in the research.
It's not necessary to describe the method in the abstract.
Introduction: same as above, the actual description of the paper's goal is introduced too late (lines 85 & following, the lines 110 &foll.). For the sake of clarity, it would be better to introduce it earlier, only announced by a very short description of the overall issue behind. E.g.: "bike sharing has evolved rapidly... Together with the obvious benefits... the growth of this business has created problems like... This paper aims to..." (please take this as an example to better explain the suggestion, this is not a mandatory scheme.). Then, after this, the reader will appreciate the rest of the introduction (previous experiences, solutions, etc.) from a different perspective, since he/she already know what the article is about.
In order to orient the article with an international audience, it is important to explain every possible contextual element whose knowledge is not spread outside the specific context of the research. This is the case, for instance, of "sesame credit", and "credit supervision system". I suggest spending a few words to clarify them.
Lines 114 & following. The actual contribution of the paper is summarised in point 1), while points 2-4) in fact describe the method. A methodological choice is a contribution if it represents a new approach to a problem that is usually investigated in a different way. In such a case, however, it would be adequate to show the novelty through a focussed literature review. Otherwise, it would be better to introduce the method in a separate section. My suggestion is to focus on the actual contribution and to extensively explain the method (in case, why it might be considered a new approach) in a specific paragraph of the paper.
Formal experimental process design. I must confess that I am a little bit puzzled by the experimental approach applied to the scope of this article. On the other hand, I am not an expert of experiments, so I would suggest that a reviewer specifically skilled in EE should express his/her opinion on this. Specifically, I cannot say whether the experiment is adequate in order to allow inferences from the differences between the groups, nor to what extent such differences are actually induced by the credit supervision mechanism.
Beyond this: line 152: "As shown in table 4, the p values of Group 1...." table 4 show something else. Table 5 and 6 show paired sample; to highlight the differences between groups it could be helpful to see descriptive for single groups. The description of results in table 5 and 6 are not clear. For instance, lines 178 and following. This is relevant since according to the authors this is the basis to state the role of the credit supervision mechanism in changing behaviors.
Section 4.3. In my opinion, this section needs to have a different articulation. First: describe the investigation as clearly and explicitly as possible. For instance: "In order to show how revealed
preferences are better predictors of behavior with respect to stated preferences, we have measured... etc." (this is just an example) Second: show the data clearly: the focus is on how revealed preferences differs from stated ones. Neither the pie diagram nor the histograms are directly showing this and figure 6 is not present. Lines 218-219: this is incomprehensible. As a whole, this section is not necessary for the basic goal of the article. I would suggest considering to remove it.
Universality of credit supervision mechanism. The regression results are not reported in the way that is commonly used in journal articles. For instance, the p-value (that SPSS name SIG) doesn't need to be reported, you could just use asterisks to highlight the threshold of significance. I suggest checking one of the many guides online about how to report regression results, for instance: http://staff.bath.ac.uk/pssiw/stats2/page2/page3/page3.html
Discussions and policy implications. One would expect to see discussed the limitations of the experimental approach in providing elements about the actual behavior, given the contextual elements that are or might be relevant in describing the actual choices. I suggest considering discussing the possible limitations of the experiment. Cnsequently, policy implications should only consider the direct implications, given the results of the research, while topics not directly related to the article scopes (e.g. role of the government / enterprises, etc.) should be dropped.
Author Response
Dear Reviewer,
In response to the referee report for “Experimental study on shared bike use behavior under the bounded rational theory and credit supervision mechanism” Manuscript ID: sustainability-390129
We would first like to thank you for the opportunity to address the reviewer’s concerns. We also appreciate the opportunity to submit a revised manuscript for consideration for possible publication in the Sustainability, Special Issue on Virtual Special Issue on Smart Technologies and Urban Life: A Behavioral and Social Perspective.
We have benefited a great deal from your comments, and we have carefully revised the paper according to the feedback received, making some substantial changes to the paper. Our responses are given in a point-by-point manner below. We hope the revised version is suitable for the special issue and look forward to hearing from you.
The detailed point-by-point response is included in the attached file. And we have also completely checked the English expression of the whole paper using official agency.
If we need to further address any of the comments, we would be happy to do so. Again, thank you.
Sincerely,
Huiyu ZHOU on behalf of all authors

Reviewer 2 Report
I like the topic of the paper and also think that the experimental setup is interesting. However, I have several remarks that may improve the paper.
I hope you find this useful.
- The paper contains many minor mistakes (e.g., typos, misplaced commas, ...). This "sloppiness" hinders reading the paper but is easy to fix. Here are a few examples just from the introduction: line 39: extra space before "and in 2018", line 48: "--", line 61: missing reference to the statista report, line 115: missing space between "2)" and "Apply", line 119: no space before (Vinayak, et al., 2017), line 123: space between "pre-" and "experimental", ...
- The literature review is interesting but feels a bit long. I'm not sure whether you need all the paper because most papers are only used in this section for positioning. Furthermore, the overview in table 1 is quite informative but the table is not linked to your literature review. In fact, the reference to Table 1 (line 86 page 2, btw I do not understand how rows and pages are counted but that's of course not the author's fault) should be a reference to table 2. Also the dimensions "Method" and "Analysis Model" are rather ambiguous. What does "Describing statistic" mean? What kind of analysis model is "Quantitative analysis" of "experimental study"?
- The experimental design is rather sophisticated, so a detailed discussion is very welcomed. However, on the one hand, some parts are still not well described (e.g., section 3.1.1. line 2: "First of all, the pre-experiment part will design questionnaires and invite participants to fill in the questionnaires, so ..." what does that mean?). On the other hand, some parts are redundant an can be shortened (e.g., in section 3.1.2. and the explanations about the cardboard and the scissors). Also, I do not understand what an incomplete sentence like "Get D point recovery, recovered by staff, put back to the original location." (line 77) should mean. Please fix and rewrite.
- Section 4: Here I'm first puzzled by the title. Instead of presenting raw numbers or descriptive results for key variables in your analysis, you start with hypotheses. Next, I don't really understand what you do to "test" your hypotheses in your "descriptive analysis". It seems like you compute all differences between the 5 groups for the variables "damage status" and "cheating". BTW: I don't understand how I should interpret the numbers in the column "mean" in tables 5 and 6. Then you test of possible differences separately and search for significant results. (Another remark: section 4.1line 152: how can a p-value be significantly lower, what does that mean?!?) This is in my view a textbook example of "multiple testing" because you are interested in a joint or global hypothesis (i.e., all mu are the same). I would recommend a better method for the analysis (i.e., a joint test like the Tukey-test) or at least the Bonferroni method to adjust the significance level (i.e., divide alpha by the number of tests).
From section 4.3. on this section indeed is a descriptive analysis, maybe the first to sections are only misplaced and can be combined with section 5? Note: the percentages in brackets are missing in the second pie-chart.
- In section 5 you perform a logistic regression. The final regression equation is not readable. Furthermore, it seems like you are just deleting non-significant terms from your model, but keeping the estimates. I think you should re-estimate the model. Also, I do not understand why some estimates in table 7 and red and in parenthesis. Also is the sign for the intercept correct, please compare the negative sign in the regression equation and the positive sign in table 7.
Author Response

(The authors gave the same response as above.)

Round 2
Reviewer 2 Report
I appreciate the effort that was spent by the authors revising the manuscript (and also providing detail responses). The paper has definitely improved; the text has gained clarity, and many minor issues have been fixed.
I'm positive about this revision but still a few problems that need to be fixed before publication:
1) I think that was (partially) a misunderstanding regarding my remark about the reference to table 1 in my last review. What I meant is very simple: Table 1 contains the literature from the review ("Related Literature Comparison"). However, you do not mention table 1 in the text as a reference. Instead, you reference to table 1 in line 433 ("as shown in Table 1"). But here you mean your table 2 ("Control and Treatment Group Settings")! So please add a correct reference to table 1 and fix the incorrect reference to table 2.
2) I appreciate that you apply now the Tukey post-hoc test, although I believe there are also better ways to test your hypothesis. Anyway, what I do not understand are specific values is table 5. Why has the difference between group 1 and 2 changed? In the first version you reported 3.778, and now you have 4.000. What happened, all other numbers (as far as I can see) stayed the same?!? Also, I think there is a typo: the SE for Group 3 vs. 5 is wrong. You report 0.086, but for the opposite comparison (5 vs. 3) it is 1.096. I think the latter is correct because of the more reasonable magnitude. Furthermore, I believe all mean differences have changed in table 6. How is this possible, you don't say anything about it in your report?!? Did you change the sample? Were there mistakes in the first round? In line 643 is also a word missing. I think you mean "significantly lower than", where lower is missing in your version.
3) I still have issues with your regression equation (line 710). Your dependent variable Y is binary; hence your equation is incorrect. The equation for a logistic regression is on the latent (linear) level. That means you compute the probability that Y = 1. P(Y = 1) = 1/(1+exp(-z)), with z = a + b1 * x1 + ... You are only showing the last part. The resulting value of your regression equation is only meaningful after the nonlinear logistic/logit transformation.
I hope this helps. All the best for your revision.
Author Response
Dear reviewer,
In response to the referee report for “Experimental study on shared bike use behavior under the bounded rational theory and credit supervision mechanism” Manuscript ID: sustainability-390129
Sincerely thank you for providing very useful and important suggestions to improve the quality of the paper and our research, we really benefited a lot through the two rounds of communication with you. The questions and suggestions are necessary, important and with keen insight. Writing the response to you also gives us another chance to deepen our understandings of the research and enriches our methodologies.
To conclude, we have benefited a great deal from your comments, and we have carefully revised the paper according to the feedback received again, making some important changes to the paper. Our responses are given in a point-by-point manner in attachment file. We hope the revised version is suitable for the special issue and look forward to hearing from you again.
If we need to further address any of the comments, we would be happy to do so. Again, thank you.
Sincerely,
Huiyu ZHOU on behalf of all authors
